# Chromatin remodeling by Pol II primes efficient Pol III transcription

Carlo Yague-Sanz[1], Valérie Migeot[1], Marc Larochelle[2], François Bachand[2], Maxime Wéry [3], Antonin Morillon [3] & Damien Hermand [1] ✉

The packaging of the genetic material into chromatin imposes the remodeling of this barrier to allow efficient transcription. RNA polymerase II activity is coupled with several histone modification complexes that enforce remodeling. How RNA polymerase III (Pol III) counteracts the inhibitory effect of chromatin is unknown. We report here a mechanism where RNA Polymerase II (Pol II) transcription is required to prime and maintain nucleosome depletion at Pol III loci and contributes to efficient Pol III recruitment upon re-initiation of growth from stationary phase in Fission yeast. The Pcr1 transcription factor participates in the recruitment of Pol II, which affects local histone occupancy through the associated SAGA complex and a Pol II phospho-S2 CTD / Mst2 pathway. These data expand the central role of Pol II in gene expression beyond mRNA synthesis.

RNA polymerase III (Pol III) is the largest of the three usual eukaryotic RNA polymerases and is specialized in the synthesis of abundant and mostly short ncRNAs named class III genes, which includes the entire repertoire of cytosolic tRNAs. The increased size of Pol III results from the fact that it contains stable integral subunits that are homologs of transcription factors in the RNA polymerase II (Pol II) system[1]. These "built-in" factors participate in all steps of the transcription cycle and partly explain the lower number of independent actors in the Pol III system. In addition, they also allow efficient recycling of the polymerase at a given locus[2]. At tRNA genes, Pol III initiates transcription at internal promoters and is recruited by TFIIIB, which itself is recruited by TFIIIC bound to A and B DNA boxes. These two elements overlap with distinctive features of the mature tRNAs. Other Pol III promoters show variations on that theme[2]. Maf1 was isolated in yeast as a global repressor of the Pol III machinery and is regulated by several signaling pathways, notably during starvation[3–5].

The idea that Pol III is a constitutive polymerase has been challenged by the finding that the relative abundance of tRNA vary across tissues[6], in response to stresses[7] and in pathologies[8]. In fission yeast, the expression level of the tRNA-ome is a key determinant of the activation of the TORC1 pathway and links nutrient availability to growth[9,10].

The high chromatin occupancy of Pol III was proposed to be sufficient to compete out the installation of a nucleosome at the target loci but the still expanding range of exquisite regulations of Pol III turns down this hypothesis and raises the question of how Pol III handles chromatin remodeling, especially upon de novo initiation seen during exit from starvation. In addition, repressing Pol III by nutritional starvation does not result in a local gain of nucleosome[11], which suggests that the remodeling of chromatin may be independent of the high level of Pol III.

About 10 years ago, three independent studies reported the presence of several typical Pol II-associated histone marks near human Pol III-transcribed genes, and Pol II was found to be enriched at class III loci[12–14]. In addition, Pol III occupancy scales with the levels of nearby Pol II but not with the levels of Pol II transcript[12], suggesting a regulatory relationship between both polymerases. This was consistent with a previous observation that Pol II facilitates Pol III transcription at the U6 gene[15]. More recently, however, another study reported that the presence of Pol II in the close proximity of class III genes is inhibitory through transcriptional interference[16], which was found to be essential for Maf1-mediated repression of a majority of tRNA genes during starvation. This repression is however limited to a small number of loci in rich media.

[1]URPHYM-GEMO, The University of Namur, rue de Bruxelles, 61, Namur 5000, Belgium. [2]RNA Group, Department of Biochemistry and Functional Genomics, Université de Sherbrooke, Sherbrooke, QC J1E 4K8, Canada. [3]ncRNA, Epigenetic and Genome Fluidity, Institut Curie, PSL Research University, Université Pierre et Marie Curie, CNRS UMR 3244 Paris, France. ✉e-mail: Damien.Hermand@unamur.be

Early genome-wide studies indicated that tRNA genes are largely devoid of nucleosomes[17], which was correlated with high RSC (Remodels Structure of Chromatin) occupancy and a gain of nucleosome density in the absence of RSC[11,18]. These data support that RSC is required to maintain class III genes in a low nucleosome density favorable to Pol III transcription. How RSC is targeted to these loci remains generally unclear although a physical interaction between Rsc4 C-terminus and Rpb5, a shared subunit of the three polymerases, was reported[19]. Apart from RSC, the ISW complexes were also implicated in the establishment of a NDR (nucleosome-depleted region) at tRNAs genes[20]. In addition, the acetylation of histone H3, typically found at class II genes, was shown to expand to tRNA genes during exponential growth[21] and two histone deacetylases, Hda1 and Hos2, are found at these loci[22], challenging the idea that tRNA genes are refractory to chromatin-based regulations. Besides the case of c-Myc recruiting TRRAP and the histone acetyl transferase GCN5 to activate Pol III in human cells[23], very little is known about how chromatin modifications and consequent remodeling occurs at class III genes.

We report here that Pol II transcription, mostly antisense, is critical to recruit the associated Gcn5 and Mst2 histone acetyltransferases to tRNAs genes respectively though SAGA and the Pol II phospho-S2 CTD. This mechanism is especially important to initiate growth from the stationary phase.

## Results

### Pol II CTD S2P is required for the maintenance of a nucleosome-depleted region and high Pol III occupancy at class three loci

Genome-wide mapping of nucleosome occupancy by MNase-seq was performed on the *rpb1* CTD S2A mutant[24,25], which revealed that the nucleosome-depleted region (NDR) found at the 171 tDNAs loci in wild-type was replaced by a distinctive peak of MNase resistance when CTD S2 phosphorylation was abolished (Fig. 1a, b). Nucleosome mapping confirmed this effect at three tDNAs loci and the non-tDNA class III gene *srp7* (Fig. 1c). Considering that these loci are transcribed by Pol III, these results were unexpected in a Pol II mutants.

To investigate this further, we used the analog-sensitive *lsk1-as* (Cdk12) allele[25] that allows the fast depletion of CTD S2 phosphorylation. Within 30 min of inhibition, the histone H3 ChIP signal increased at most Pol III target loci tested without affecting the class II *act1* locus (Fig. 1d). These data indicate that Pol II CTD phosphorylation on S2 is required to counteract the formation of an H3-containing structure, likely a nucleosome, at class III loci and therefore to maintain an NDR at class III loci. Importantly, the occupancy of Pol II at protein-coding genes or tDNA was barely affected in the *rpb1 CTD S2A* mutant (Fig. 1e). In addition, the effect was specific to S2 as the inactivation of the CTD S5 kinase Mcs6 (CDK7) did not increase H3 occupancy at the tDNA loci tested (Fig. 1f).

Rpc1-TAP (Rpc1 encodes the largest subunit of Pol III) ChIP-seq revealed that the level of Pol III associated with chromatin was strongly reduced in the *rpb1* CTD S2A strain (Fig. 2a) for 117 out of 171 tDNA loci while this effect was not systematic for the subset of tDNA clustered within the centromeric repeats (Fig. 2b). The fast inhibition of the CTD S2 kinase Lsk1 also quickly reduced the occupancy of Pol III (Fig. 2c). A hypothesis to explain this effect is that the loss of the NDR resulting from lower CTD S2P impedes the occupancy of Pol III. However, the loss of the NDR may also be a consequence of the lower chromatin association of Pol III when Pol II CTD S2P is abolished.

### CTD-phosphorylated RNA Pol II transcribes class III loci

To test if Pol II may directly affect class III genes by transcribing these regions, we analyzed our previous ChIP-Seq data of the phosphorylated Pol II[26]. Strikingly, we found that the CTD-phosphorylated Pol II associates with virtually all tDNA loci and the Pol II peak is generally centered on the tRNA TSS (Fig. 3a). A closer analysis of the patterns of CTD S2P and S5P revealed that the S2P peak is slightly shifted upstream

of the orientation of the tDNA gene compared to the almost overlayed S5P and Pol II peaks (Fig. 3b), which may result from a mostly antisense directionality of Pol II at these Pol III loci. Also supporting this possibility, our previous Pol II NET-Seq profiles[27] revealed the low abundance of nascent Pol II-associated transcripts mapping antisense to tRNA (Fig. 3c, d). Strand-specific RT-qPCR confirmed the presence of an Exosome-sensitive antisense RNA to the *ARG.05* and *ILE.04* tRNAs (Supplementary Fig. 1A), which was confirmed globally by metagene analysis (Supplementary Fig. 1B). We also found a small ($R^2 = 0.45$), yet significant correlation between the level of S2P and the level of antisense transcription detected by NET-Seq (Supplementary Fig. 1C). Finally, independent ChIP experiments confirmed that CTD S2 phosphorylated Pol II is present at tDNA loci and revealed that the rapid inhibition of S2P results in a decreased occupancy of Pol II of about 50% (Fig. 3e and Supplementary Fig. 1D). As we did not find a significant correlation between the level of Pol III and Pol II, Pol II transcription may not be limiting for efficient Pol III occupancy, at least in exponential growth.

### Reducing Pol III occupancy does not affect the histone level at tRNA loci

While in the process of setting up a Pol III ChIP assay, we first tagged the Rpc25 subunit with a flag. Contrary to Rpc1-TAP, the Rpc25-flag strain grew poorly, especially at 37 °C (Supplementary Fig. 2A) and had a reduced level of chromatin-bound Pol III (Supplementary Fig. 2B) while the overall level of Rpc1 was not affected (Supplementary Fig. 2C).

Previous work in *S. cerevisiae* highlighted the importance of the well-conserved and essential Rpc25 subunit for initiation[28]. Structural data support that a C-terminal tag of Rpc25 impedes the conformational change required for the formation of pre-initiation complex (Supplementary Fig. 2D). The *rpc25-flag* allele is therefore hypomorphic and was useful to study the connection between Pol II and Pol III transcription at tDNA loci. Notably, ChIP analyses showed that the reduced occupancy of Pol III in the *rpc25-flag* strain is not correlated with a higher histone occupancy (Supplementary Fig. 2E), indicating that the lower occupancy of Pol III seen in the *rpb1* CTD S2A mutant (Fig. 2a) is unlikely to directly cause the increased nucleosome occupancy observed. In addition, an earlier work showed that repressing Pol III by nutritional starvation does not result in a local gain of nucleosome[11], supporting this conclusion. Nevertheless, in the case of the *SNR6* gene, the upstream nucleosome, which covers the TATA box under repressed conditions is shifted -50 bp further upstream by RSC upon activation[29].

Taken together, these results support that Pol II transcription of class III loci is required for the maintenance of a NDR and resulting high level of Pol III at tDNAs.

In budding yeast and higher eukaryotes, active transcription of tRNAs is globally correlated with Pol III occupancy on tDNAs chromatin[30,31]. Therefore, we expected that the lower occupancy of Pol III in the *rpb1* CTD S2A mutant should result in lower tRNAs (or pre-tRNAs) steady-state despite the fact that exponential growth is not affected in the *rpb1* CTD S2A mutant[32]. RNA-seq revealed however that the level of most of the 171 mature tRNAs is not decreased in the *rpb1 CTD S2A* mutant while their precursors are slightly increased (Supplementary Fig. 3). These data suggest that a post-transcriptional buffering mechanism is active during log-phase growth as previously reported in various other cases of transcriptional downregulation[33,34]. Considering the large amount of sense transcripts stabilized in the *rrp6* mutant (Supplementary Fig. 1B), it could be that this buffering mechanism is mediated by the Exosome. Further work is necessary to explore this possibility.

### Pcr1, SAGA, and Mst2 are required for efficient Pol III chromatin occupancy

Pcr1 is a bZIP transcription factor homologous to CREB and binding the CRE motif TGACGTCA[35] to recruit Pol II. Pcr1 was found associated with

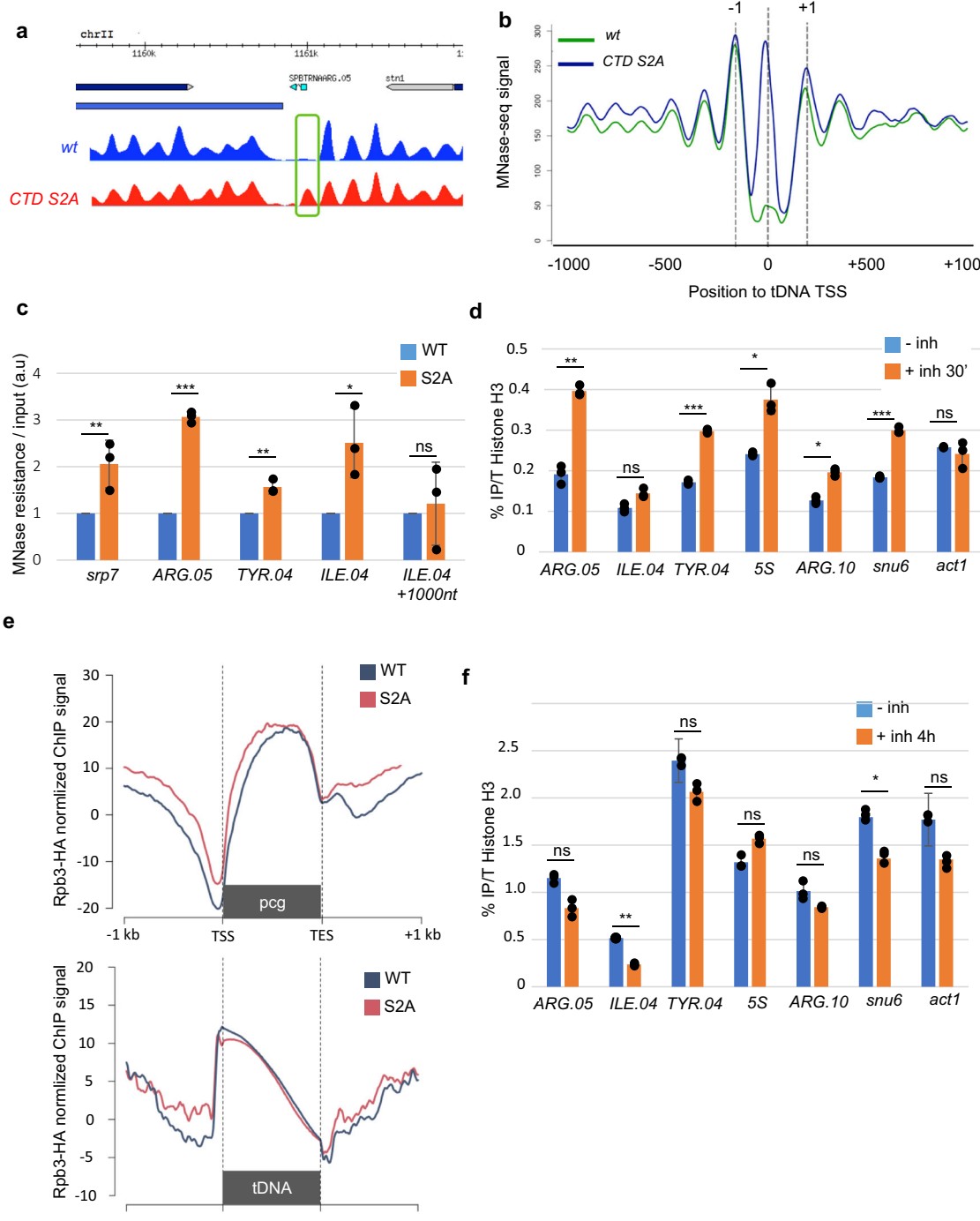

**Fig. 1 | Pol II CTD S2P is required for the maintenance of a nucleosome-depleted region at class three loci. a** Snapshot of the MNase-seq profiles[25] in the indicated strains at the *ARG.05* tDNA loci. **b** Metagene average profile of MNase-seq signal[25] in the indicated strains centered on tRNAs TSS (*n* = 171) and expanding over 1- kb flanking regions. The +1 and −1 nucleosome are indicated. **c** Nucleosome scanning/ MNase-qPCR in the indicated strains at the indicated class III loci. The *ILE.04* +1000 nt locus refers to an intergenic region located 1000 nt upstream of ILE.04. Each column represents the averaged value and error bars are the standard deviation (*n* = 3 biological replicates), *P < 0.05, **P < 0.01, ***P < 0.001; ns not significant upon paired *t* test. **d** ChIP experiment performed on chromatin prepared from the Lsk1-as strain grown in the absence (− inh) or in the presence (+ inh 30') of 10 μmol 3-MB-PP1 inhibitor using anti-H3 antibody. The amplicons targeted the indicated

class III loci. Each column represents the averaged value and error bars are the standard deviation (*n* = 3 biological replicates), *P < 0.05, **P < 0.01, ***P < 0.001; ns not significant upon paired *t* test. **e** Metagene of normalized Rpb3-HA ChIP signal around all non-convergent protein-coding genes (pcg, top panel, *n* = 3520) and tRNA genes (tDNA, bottom panel, *n* = 171) in wt and the *rpb1 CTD S2A* mutant. Normalization was performed on spiked-in chromatin from a *S. cerevisiae* Rpb3-HA strain. **f** ChIP experiment performed on chromatin prepared from the Mcs6−as strain grown in the absence (− inh) or in the presence (+ inh 4 h) of 30 μmol 3-MB-PP1 inhibitor using anti-H3 antibody. Each column represents the averaged value and error bars are the standard deviation (*n* = 3 biological replicates), *P < 0.05, **P < 0.01, ***P < 0.001; ns not significant upon paired *t* test. Source data are provided as a Source Data file.

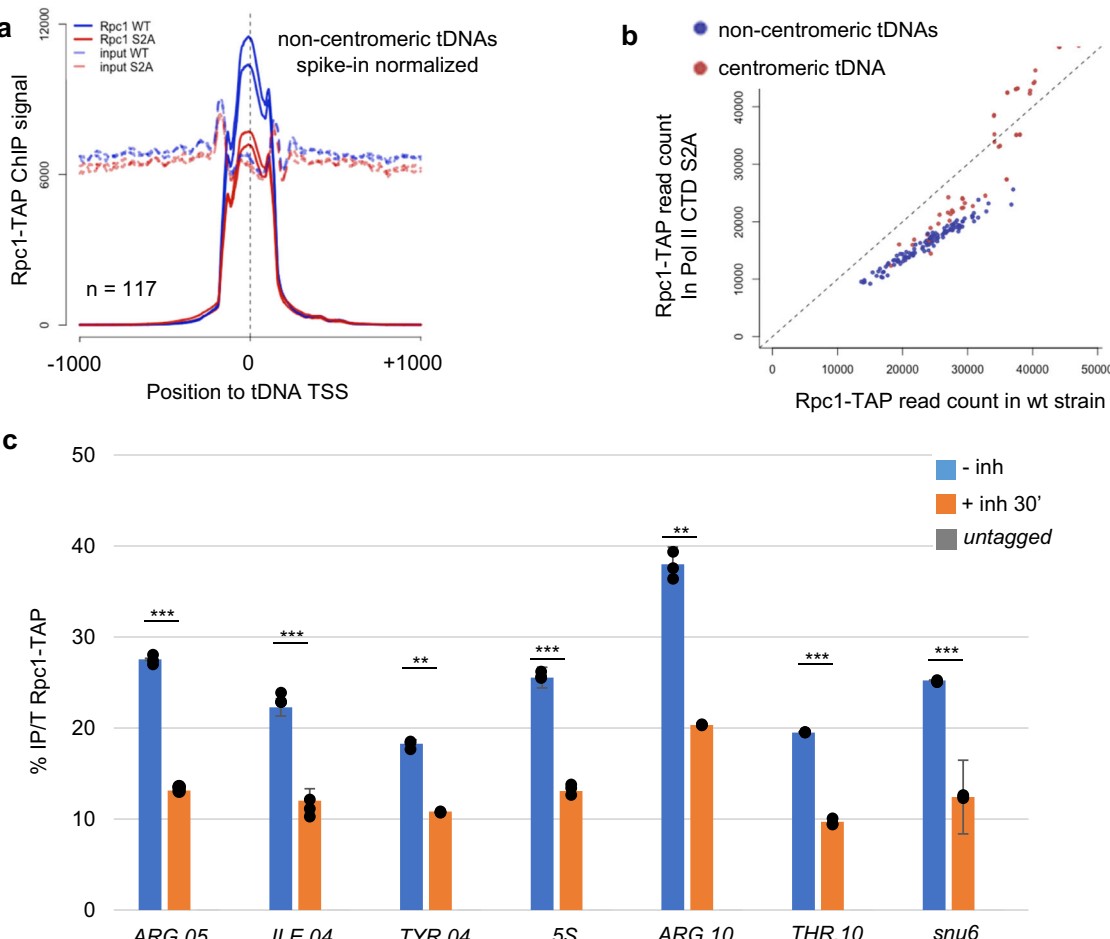

**Fig. 2 | Pol II CTD S2P is required for high Pol III occupancy at class three loci.**
**a** Metagene average profile of Rpc1-TAP ChIP-seq or input signal centered on tRNAs TSS ($n = 117$) and expanding over 1 kb flanking regions. The signal was normalized on the total number of spike-in reads by condition. **b** Scatter plot of the Rpc1-TAP peak intensity on tDNAs in the indicated strains summed over two replicates and normalized on the total number of spike-in reads by condition. Centromeric and non-centromeric tDNAs are distinguished by the color code. **c** ChIP experiment performed on chromatin prepared from the Rpc1-TAP Lsk1-as strain grown in the absence (− inh) or in the presence (+ inh 30′) of 10 µmol 3-MB-PP1 inhibitor and an untagged control using anti-TAP antibody. The amplicons targeted the indicated class III loci. Each column represents the averaged value and error bars are the standard deviation ($n = 3$ biological replicates), *$P < 0.05$, **$P < 0.01$, ***$P < 0.001$; ns not significant upon paired $t$ test. Source data are provided as a Source Data file.

class III loci in a previous genome-wide study[36] and a genuine CRE is found within 1 kb downstream of 30% of the tDNA loci in *S. pombe*. In addition, the overexpression of the Pcr1 partner Atf1 results in the overexpression of about one-fourth of the fission yeast tRNAs[36].

Based on these converging preliminary data, we tested if the deletion of *pcr1* affects the occupancy of Pol II at class III loci. The level of Pol II was reduced at tDNAs when Pcr1 was absent (Fig. 4a). ChIP-seq revealed that the level of Pol II was decreased at about 44% (75 out of 171) of the tDNAs loci when the *pcr1* gene was deleted (Fig. 4a). In addition, the absence of Pcr1, resulted in an increase of the level of H3 and a decrease of Rpc1-TAP, which is reminiscent of what we observed in the Pol II CTD S2A mutants.

Pcr1 was previously reported to recruit the SAGA complex[37]. SAGA (Spt-Ada-Gcn5 acetyltransferase) is an evolutionarily conserved general transcriptional coactivator of Pol II notably containing a HAT (Histone acetylation) module harboring the Gcn5 acetylase[38,39]. ChIP experiments revealed that Gcn5 is enriched at class III genes and that its deletion locally affects histone H3 acetylation (Fig. 4b).

While there is no direct link between SAGA and the Pol II S2P CTD, the Set2 histone methyltransferase was reported to be a downstream effector of the CTD S2P mark[40]. While we failed to efficiently chip Set2 at class III loci, the deletion of *set2* increased the level of H3 (Supplementary Fig. 4A) and strongly affected the occupancy of the Mst2 HAT (Fig. 5a), which is one of its downstream effectors[41]. Confirming the result, the mutation of the SRI domain of Set2, known to specifically abolish the deposition of H3K36me3[41], was sufficient to abolish the chromatin recruitment of Mst2 (Fig. 5a).

An emerging model from these data is that transcribing Pol II at class III genes independently recruits two of its associated HAT activities through both its coactivator SAGA (Gcn5) and a CTD S2P-Set2[H3K36me3]-Mst2 branch. Because Mst2 was reported to function with Gcn5 to regulate the global level of H3K14ac[42], we next tested the effect of a combined inactivation of the two HAT, which synergistically affected the occupancy and the acetylation of H3 (Supplementary Figs. 5B and 4B). The chromatin level of Rpc1-TAP was reduced in the double mutant (Supplementary Fig. 4C).

**RNA Pol II-associated chromatin remodeling is critical for efficient Pol III recruitment upon reinitiation of growth from stationary phase**

We previously reported that the *rpb1 CTD S2A* mutant has a "slow start" phenotype upon germination of spores from an heterozygous diploid but then reached a midlogarithmic phase doubling time similar to that of wild-type[32]. These data suggested that the mutant has difficulties to re-establish growth upon exit from the stationary phase[32]. Exit from

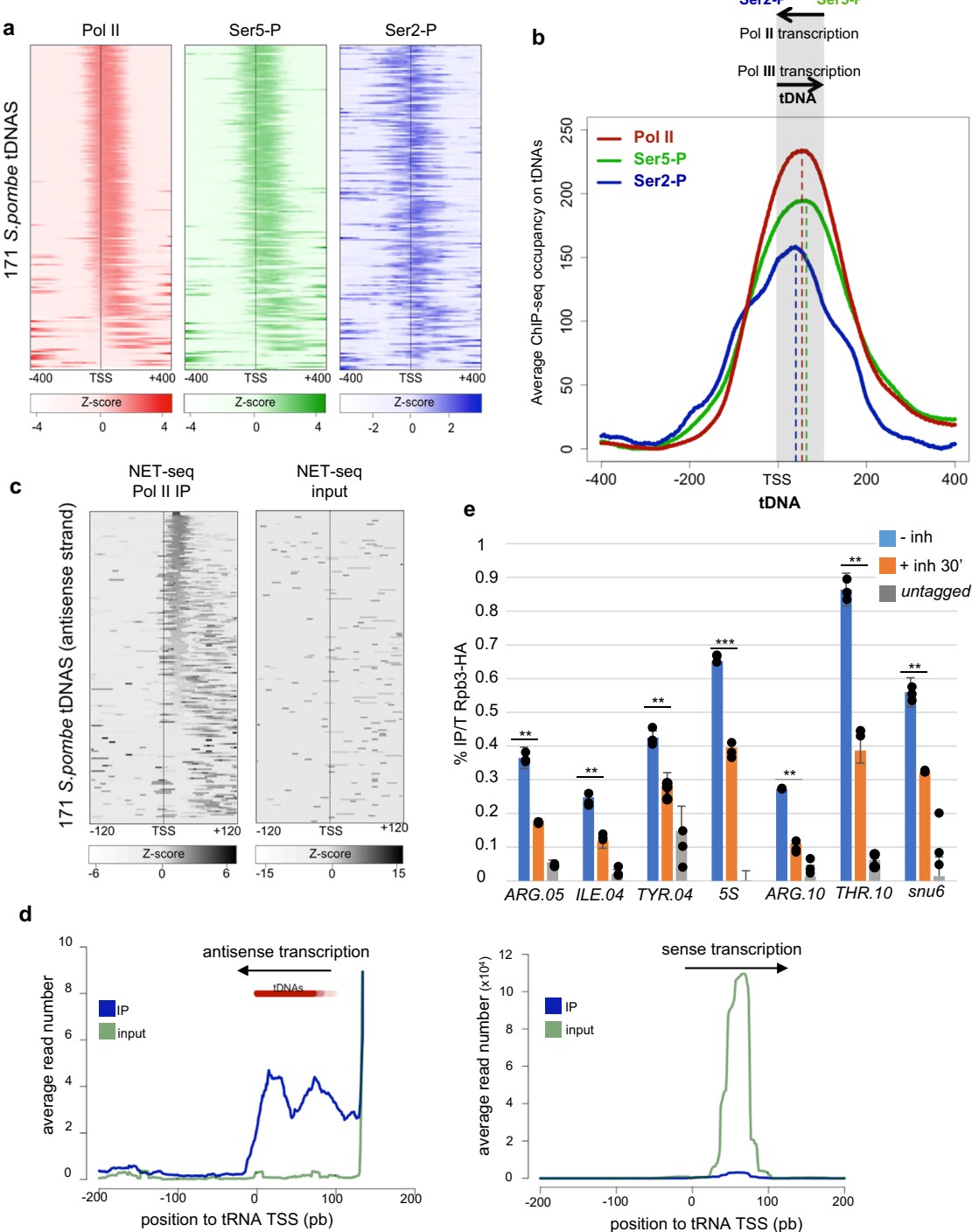

**Fig. 3 | CTD-phosphorylated RNA Pol II transcribes class III loci antisense to Pol III. a** Heatmaps of the Z-score of the ChIP signal for Pol II (red), CTD S2P (green) and CTD S5P (blue) centered on tDNA TSS (*n* = 171). The raw data were from our previous work[26]. The box indicates the lower (25%) and upper (75%) quartile and the black line indicates the median. Whiskers extend to the most extreme data points. **b** Metagene average profile of ChIP-seq signal for Pol II (red), CTD S2P (green) and CTD S5P (blue) centered on tDNAs TSS (*n* = 171). The raw data were from our previous work[26]. **c** Heatmaps of the Z-score of the NET-seq signal on the antisense strand for Pol II IP or input centered on tDNA TSS (*n* = 171). The raw data were from our previous work[27]. **d** Metagene analysis of the sense (left) and antisense (right)

NET-seq data (IP and input as indicated) covering a −200 bp to +200 bp region centered to the tRNA transcription start site (0). Note that the strong increase observed at about +130 bp in the antisense panel corresponds to clustered tRNAs located on the complementary strand. **e** ChIP experiment performed on chromatin prepared from the Rpb3-HA Lsk1-as strain grown in the absence (− inh) or in the presence (+ inh 30′) of 10 μmol 3-MB-PP1 inhibitor and an untagged control using anti-HA antibody. The amplicons targeted the indicated class III loci. Each column represents the averaged value and error bars are the standard deviation (*n* = 3 biological replicates), *\**P* < 0.05, **\**P* < 0.01, ***\**P* < 0.001; ns not significant upon paired *t* test. Source data are provided as a Source Data file.

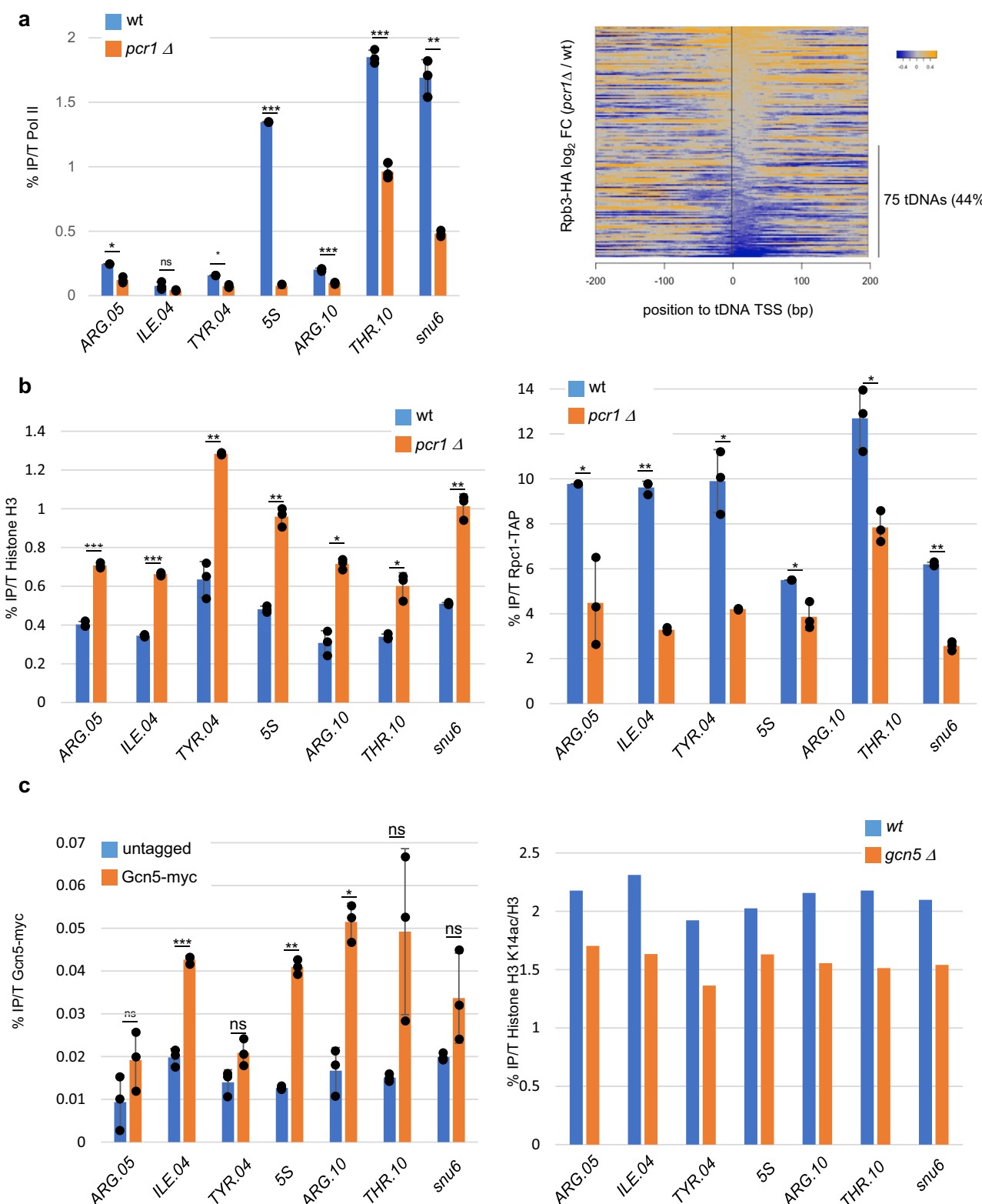

stationary phase correlates with a marked recruitment of Pol III at class III loci, which was reduced when Rpb1 CTD S2 cannot be phosphorylated (Fig. 6a). A double *gcn5 mst2* mutant was also defective at increasing the chromatin association of Rpc1 upon exit from starvation (Fig. 6a). The phosphorylation of the CTD on S2 and the acetylation of H3 were required to efficiently initiate growth from stationary phase but this requirement was much weaker when cells were maintained in exponential phase (Fig. 6b and Supplementary Fig. 5A).

Taken together, these data support that a normal Pol II cycle and its associated chromatin modification/remodeling complexes are required to quickly restore a high level of Pol III at class III genes and ensure efficient growth.

In order to test this model, we introduced the well-characterized and strong *ura4* terminator sequence[43] downstream of the *PRO.08* tDNA loci to block antisense Pol II transcription initiated downstream of the tRNA. The construct was designed to also remove the putative

**Fig. 4 | SAGA-associated chromatin remodeling is required for efficient Pol III transcription. a** Left panel: ChIP experiment performed on chromatin prepared from a Rpc1-TAP *pcr1Δ* and a Rpc1-TAP control using anti-Pol II (8WG16) antibody. The amplicons targeted the indicated class III loci. Each column represents the averaged value and error bars are the standard deviation ($n = 3$ biological replicates), *$P < 0.05$, **$P < 0.01$, ***$P < 0.001$; ns not significant upon paired $t$ test. Right panel: Heatmap of Rpb3-HA ChIP-seq signal centered on tRNAs TSS ($n = 171$) and expanding over 200 bp flanking regions in a *pcr1* deletion strain compared to wt. The 75 tDNA loci showing at least 25% decreased level of Pol II are indicated. **b** ChIP experiment performed on chromatin prepared from a Rpc1-TAP *pcr1Δ* and a Rpc1-TAP control using anti-histone H3 (left panel) or anti-TAP (right panel). The amplicons targeted the indicated class III loci. Each column represents the averaged value and error bars are the standard deviation ($n = 3$ biological replicates), *$P < 0.05$, **$P < 0.01$, ***$P < 0.001$; ns not significant upon paired $t$ test. **c** Left panel: ChIP experiment performed on chromatin prepared from a Gcn5-myc and an untagged control using anti-myc antibody. The amplicons targeted the indicated class III loci. Each column represents the averaged value ± standard deviation ($n = 3$ biological replicates). Right panel: ChIP experiment performed on chromatin prepared from a *gcn5 Δ* and a *wt* control using anti-H3 and anti-H3K14ac antibodies (same chromatin sample). The H3K14ac signal was normalized on the H3 signal. Each column represents the averaged value and error bars are the standard deviation ($n = 3$ biological replicates), *$P < 0.05$, **$P < 0.01$, ***$P < 0.001$; ns not significant upon paired $t$ test. Source data, including the unnormalized data used to make the right panel, are provided as a Source Data file.

CRE motif located 550 bp downstream of the *PRO.08* tDNA. Indeed, NET-seq and anti-Rpb3, anti-CTD S2P and anti-CTD S5P ChIPs all support that *PRO.08* tDNA antisense Pol II transcription is initiated from that site (Fig. 6c). Importantly, the expression of the downstream Pol II gene *SPBC11C11.06c* is dependent on neither Pcr1 nor Atf1[44].

Blocking antisense Pol II transcription (Fig. 6d, top panel) was sufficient to impede the recruitment of Rpc1 at the *PRO.08* loci upon exit from the stationary phase without affecting the control *PHE.03* loci (Fig. 6d, bottom panel). These data support that Pol II plays a critical role in the efficient recruitment of Pol III upon exit from the stationary phase and reinitiation of growth.

## Discussion

Contrary to Pol II, the Pol III system has no known gene-specific regulators, yet the relative abundance of individual tRNAs varies extensively across cell lines and conditions[7]. It was also noted that about 20% of the active tRNA genes reside within 1 kb of the transcription start site (TSS) of a Pol II transcribed gene and are decorated by most of the active chromatin marks classically associated with Pol II transcription. Intriguingly, the class III genes that are not in the proximity of Pol II TSS also have histone modifications found at Pol II transcribed loci and are bound by Pol II, suggesting that Pol II transcription, sometimes referred to as "pervasive" occurs at these loci independently of a neighbor class II gene[12–14]. These data raised hypotheses about the putative regulatory role of Pol II transcription at class III loci.

A recent study established that Pol II is a direct regulator of tRNA transcription by robustly interfering with Pol III function. The effect, which is limited to a small number of loci in rich media conditions becomes essential for MAF1-repression upon serum starvation[16].

We report here that Pol II, clearly including tDNA antisense transcription, is required for chromatin remodeling at class III genes. We propose that Pol II transcription directly alters the chromatin template through at least two of its histone acetylation-associated activities, namely the Gcn5-containing SAGA complex and Mst2. We show that these two HATs synergize to maintain high acetylation and low histone occupancy at class III genes, therefore maintaining high level of chromatin-bound Pol III (Fig. 7).

SAGA is a conserved multi-subunit coactivator recruited at promoters[38] by gene-specific transcription factors including Pcr1 in fission yeast[37]. The Mst2 HAT was shown to be sequestered at transcriptionally active genes by the H3K36me3 mark[41]. This mark is generated by the Set2 enzyme, which is itself recruited by the phospho-S2 CTD as a consequence of Pol II transcriptional activity[40,45]. Therefore, Mst2 is directly dependent upon the phosphorylated Pol II.

The requirement of Pol II for efficient Pol III transcription may be especially important upon exit from the stationary phase when cellular growth must be quickly reactivated.

We do not consider the above model and the one proposed by Gerber et al. contradictory or mutually exclusive because the latter is specifically relevant in starvation. Therefore, a possibility is that upon starvation, Pol II transcription is increased at class III loci and interferes with sense Pol III, which facilitates Maf1-dependent inhibition. Moreover, our data are in agreement of previous data reporting the close association of RNA polymerase II and several transcription factors with class III loci and their requirement for efficient expression of Pol III genes[46]. The experiments presented here brings a mechanistic framework for these early observations.

At about 44% of the tDNA loci, the Pol II transcription we detected requires the Pcr1 transcription factor that was shown to be enriched at tDNA genes[36]. Other transcription factors, including Gaf1 that often occupy these regions, may participate in the recruitment of Pol II[47]. While a clear CRE binding motif is detected is detected within 1 kb of 44 tRNA genes, Pol II transcription may also originate from neighbor class II genes, which may explain the more disperse Pol II ChIP and NET-seq signals seen at some tDNA loci (Fig. 3a, c). The fact that Pcr1 is a target of MAP kinase signaling activated by stress conditions[48] is interesting in that context. Further work is needed to test the possibility that upon starvation, a Pcr1-dependent increase of Pol II transcription interferes with Pol III similarly to human cells.

We cannot formally exclude that the strong impact on tDNA NDR we observed is a secondary effect of interfering with Pol II transcription that would affect the transcription of some chromatin regulators. However, this possibility is unlikely as the effect we report is nearly exclusively specific to class III loci. In addition, the analysis of the transcriptome of the CTD S2A mutant did not reveal a significant downregulation of any chromatin regulators[32] or any global perturbation of the chromatin landscape. Finally, the inhibition of the CTD S2 kinase Lsk1 affects H3 occupancy at tDNA within 30 min.

The present work shows that the presence of the Pol II machinery that was detected at class III loci in various species plays an important role in local chromatin remodeling, which is required for efficient Pol III recruitment and activity.

## Methods
### Fission yeast methods
*Schizosaccharomyces pombe* strains were grown in YES (Formedium PCM0310) or EMM (MP 4110-012) at 32 °C to $OD_{595nm}$ 0.5–0.8. For spot assays, strains were cultured in liquid YES media at 32 °C and harvested at exponential phase, spotted (tenfold dilutions) on YES agar plates, and incubated up to 3 days at 32 °C[49].

For transgenesis in *S. pombe*, Expand High Fidelity-generated PCR products (Roche 19012322) were generated by amplification of a nucleotide sequence (consisting of a resistance cassette against an antibiotic or a tag sequence cloned in a pFA6A vector) with primers designed by the Pombe PCR Primer Program (developed by the Bähler lab, http://www.bahlerlab.info/resources/, 3'-end 20 nt homology with the vector and 5'-end 60 nt homology with the genome). Purified PCR products (Qiagen 28106) were transformed by following the lithium acetate procedure[50]. Gene deletions resulted in ATG-STOP replacement and gene tagging resulted in a C-terminal tagging by removal of the endogenous STOP codon and insertion of the tag sequence.

For the experiment of Fig. 6, strains were gown to stationary phase for 17 h until OD 4 (stat), diluted in fresh medium to OD 0.3, and grown for 1 h (log).

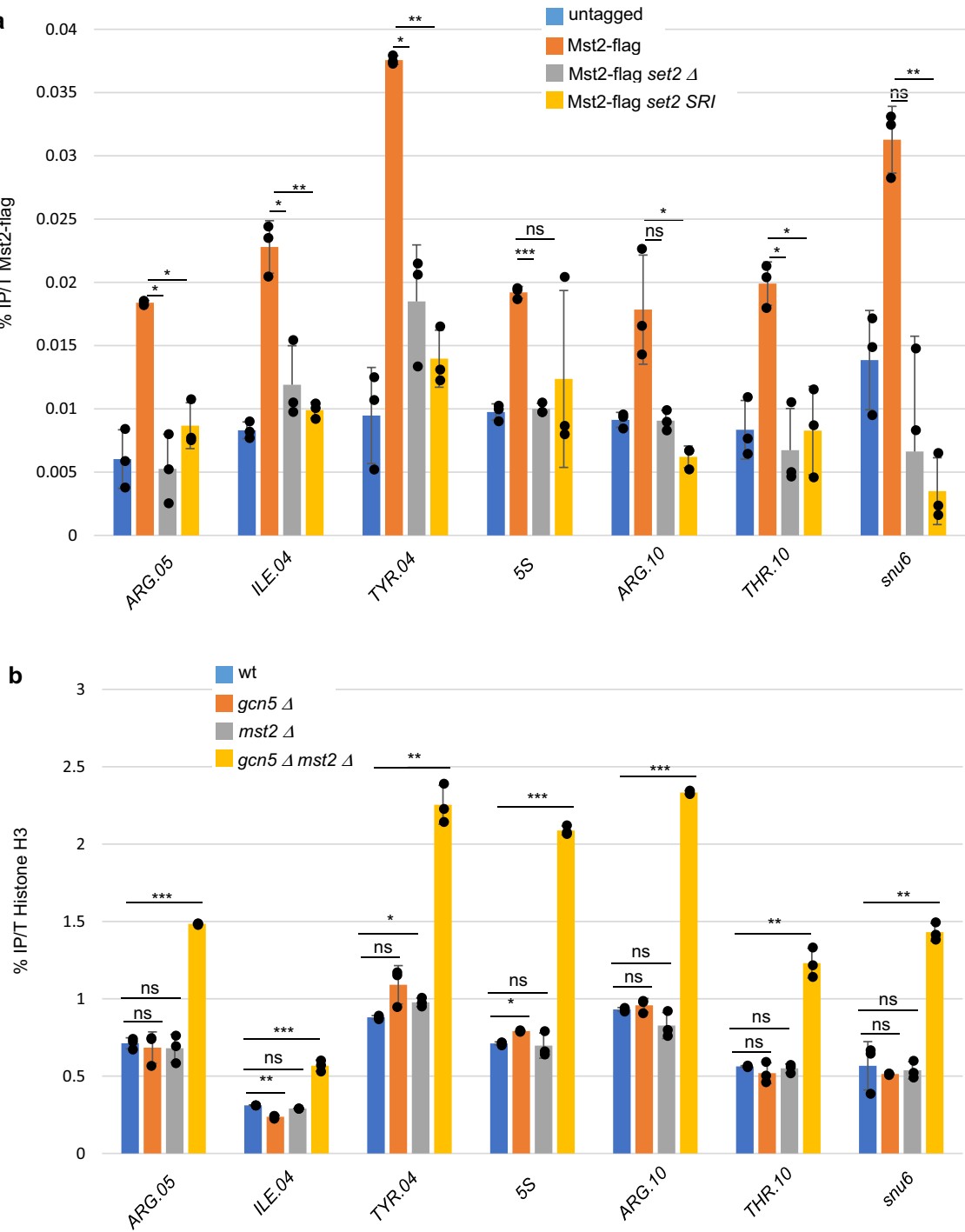

**Fig. 5 | The Mst2-associated histone acetylation is required for efficient Pol III transcription. a** ChIP experiment performed on chromatin prepared from the indicated strains using anti-flag antibody. The amplicons targeted the indicated class III loci. Each column represents the averaged value and error bars are the standard deviation ($n = 3$ biological replicates), $*P < 0.05$, $**P < 0.01$, $***P < 0.001$; ns not significant upon paired $t$ test. **b** ChIP experiment performed on chromatin prepared from the indicated strains using anti-H3 antibody. The amplicons targeted the indicated class III loci. Each column represents the averaged value and error bars are the standard deviation ($n = 3$ biological replicates), $*P < 0.05$, $**P < 0.01$, $***P < 0.001$; ns not significant upon paired $t$ test. Source data are provided as a Source Data file.

The *PRO.08 ura4T* strain was constructed by integrating the strong *ura4* terminator (defined in ref. 43) downstream of the *SPBTRNAPRO.08* loci. The integration removes 452 bp next to the poly-T Pol III termination signal and deletes the TGACGTC CRE motif. Growth was determined based on optical densities measured at 595 nm on a BIOTEK Absorbance microplate reader Epoch2NS according to the instructions of the manufacturer. The yeast strains

and primers for qPCR and ChIP described in this study are listed in Supplementary Data 1.

### Chromatin immunoprecipitation (ChIP)
Chromatin Immunoprecipitations were performed using a Bioruptor (Diagenode) and Dynabeads (Invitrogen)[49]. Briefly, cells were grown in YES medium at 32 °C until late/mid-exponential phase (OD$_{600}$ ~0.8).

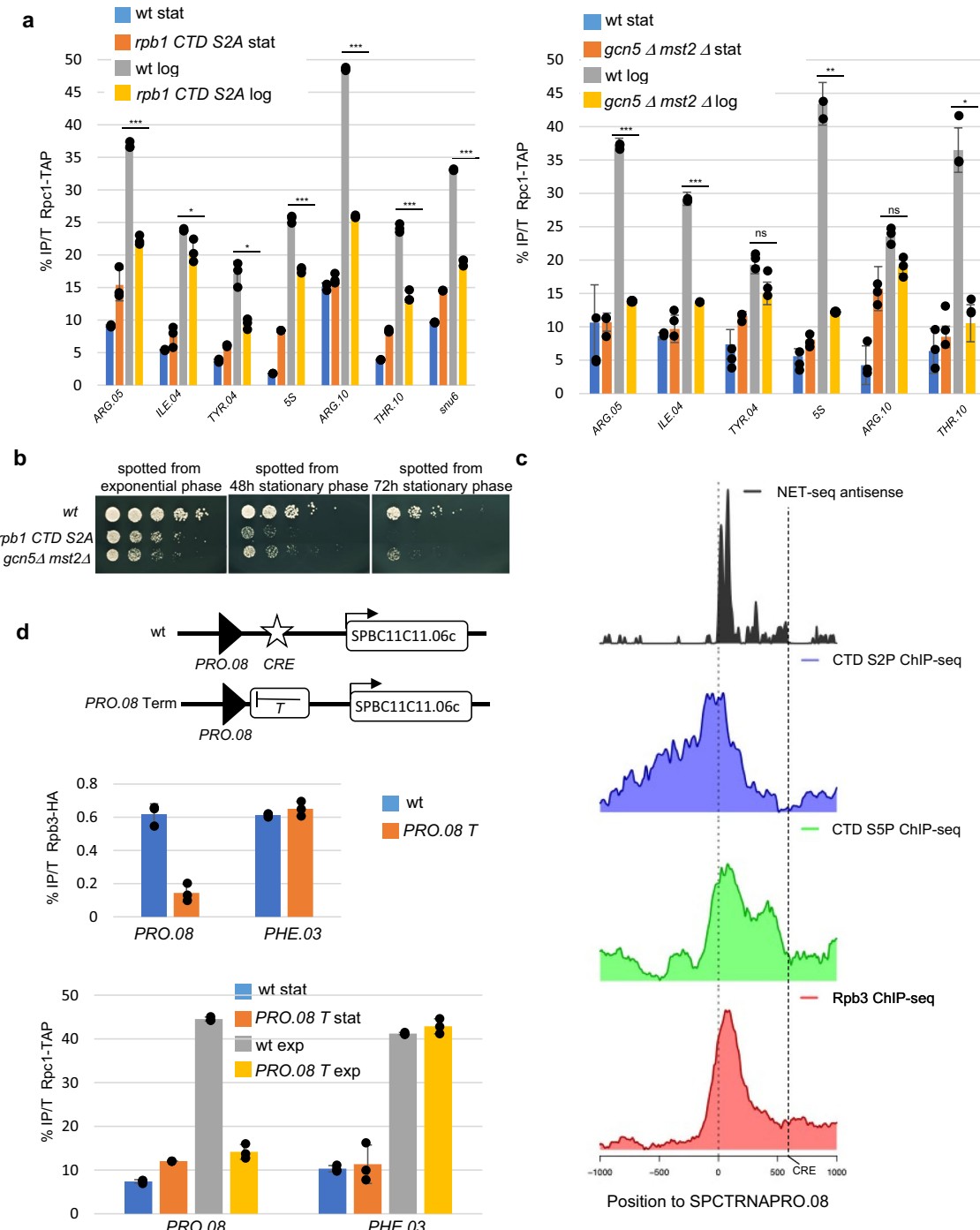

**Fig. 6 | RNA Pol II-associated chromatin remodeling is critical upon reinitiation of growth from the stationary phase. a** Left panel: Rpc1-TAP ChIP experiment performed on chromatin prepared from strains with the indicated *rpb1* backgrounds grown to stationary phase (stat) or after 1 h after reinitiation of growth following dilution (log). The amplicons targeted the indicated class III loci. Each column represents the averaged value and error bars are the standard deviation (*n* = 3 biological replicates), \*P < 0.05, \*\*P < 0.01, \*\*\*P < 0.001; ns not significant upon paired *t* test. Right panel: Rpc1-TAP ChIP experiment performed on chromatin prepared from strains with the indicated *gcn5 mst2* backgrounds grown to stationary phase (stat) or after 1 h after reinitiation of growth following dilution (log). The amplicons targeted the indicated class III loci. Each column represents the averaged value and error bars are the standard deviation (*n* = 3 biological replicates), \*P < 0.05, \*\*P < 0.01, \*\*\*P < 0.001; ns not significant upon paired *t* test. **b** Growth assay at 32 °C of the indicated strains spotted from exponential growth phase cultures (left) or from cultures maintained in stationary phase (right) for 48

or 72 h. **c** NET-seq, anti-Rpb3, anti-CTD S2P, and anti-CTD S5P profiles at the *PRO.08* tDNA locus centered on the tRNA TSS. The putative CRE site is indicated. **d** Top panel: A schematic of the *PRO.08* loci. "CRE" indicates the CRE motif located 478 bp downstream of the tRNA and "T" indicates the *ura4* terminator. Middle panel: Rpb3-HA ChIP experiment performed on chromatin prepared from a wt strain (wt) or a strain with a terminator blocking Pol II transcription (*PRO.08 T*). The amplicons targeted the indicated *PRO.08* or *PHE.03* loci. Each column represents the averaged value and error bars are the standard deviation (*n* = 3 biological replicates). Bottom panel: Rpc1-TAP ChIP experiment performed on chromatin prepared from a wt strain (wt) or a strain with a terminator blocking Pol II transcription (*PRO.08 T*) grown to stationary phase (stat) or after 1 h after reinitiation of growth following dilution (log). The amplicons targeted the indicated *PRO.08* or *PHE.03* loci. Each column represents the averaged value and error bars are the standard deviation (*n* = 3 biological replicates). Source data are provided as a Source Data file.

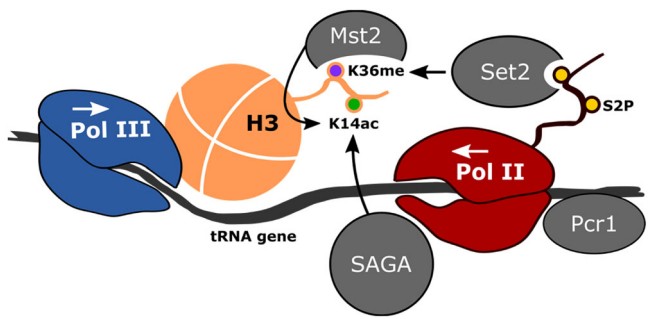

**Fig. 7 | A model of Pol II-dependent chromatin remodeling at class III loci.** In this example, Pcr1 transcription factor is required to recruit Pol II and its associated co-factor SAGA that harbors the Gcn5 HAT to allow tDNA antisense transcription. The Pol II Phospho-S2 CTD results in H3K36 methylation that recruits the second HAT Mst2. Variations on that model including Pol II transcription emerging from a close class II gene are not shown.

Then 80 mL of cultures were incubated for 10 min at room temperature with 1% formaldehyde. After quenching the reaction with glycine, cells were washed in cold Tris-buffered saline (20 mM Tris-HCl pH 7.5, 150 mM NaCl) and snap-frozen. The pellet was thawed on ice and washed with FA/SDS/PI (HEPES-KOH pH 7.5 50 mM, NaCl 150 mM, EDTA 1 mM, Na deoxycholate 0.1%, Triton X-100 1%, SDS 0.1%, PMSF 1 mM), resuspended in 1.6 ml of FA/SDS/PI and incubated 15 min on ice. The sample was sonicated with a BioRuptor (Diagenode) (cycles of 30 s ON, 60 s OFF, 21 min). After transfer to 1.5 ml microcentrifuge tube and centrifugation for 30 min, 14,000 × *g*, 4 °C, the supernatant was transferred to a new tube. Per IP, 50 μl of appropriate magnetic beads were washed four times with PBS BSA (1 mg/ml) and resuspended in PBS BSA (10 mg/ml) and then added to each 500 μL chromatin extract IP and incubated 2 h at 21 °C. The beads were washed three times with 1 ml of FA/SDS+NaCl 500 mM, one time with 500 μl of IP Buffer and one time with 500 μl of TE and resuspended in 125 μl of Pronase Buffer 1× and incubated 20 min at 65 °C. RNAse A was added 1 h at 37 °C and the DNA sample were purified on Rapace columns (Qiagen).

Antibodies used in ChIP were anti-Pol II (Covance #MMS-126R), anti-H3 (Abcam #1791), anti-H3 K14-ac (Millipore #07-353), anti-HA (Sigma #H6908), PAP (Sigma #P1291), anti-CTD S2P (Millipore #04-1571), anti-CTD S5P (Abcam ab5408), anti-flag (Sigma #F3165) and anti-Myc (Covance #MMS-150P). For all ChIP experiments, each column represents the mean percentage immunoprecipitation value ± standard deviation. The number of biological replicates is indicated in the legend (*n*). Primers are listed in Supplementary Data 1.

## MNase-seq and MNase-ChIP

The MNase-seq datasets and MNase-ChIP protocol were reported in refs. 25,51. Shortly, mononucleosomal DNA for sequencing was isolated[25,51]. The amount of Zymolyase 20T used to prepare spheroplasts was optimized experimentally for each *S. pombe* strain and for the different physiological conditions to generate a 80:20 ratio of mononucleosomes to dinucleosomes. Mononucleosomal DNA isolated as described above was sequenced in an Illumina Genome Analyzer IIx. 16588557 to 35703552 single reads 36 or 40 nucleotides long, depending on the sample, representing an average genome coverage ranging from 46- to 177-fold were aligned to the *S. pombe* reference genome. The alignment of reads generated two peaks (one on each strand) corresponding to the boundaries of each nucleosome. We used the smoothed signal generated by using the multilevel 1-D biorthogonal wavelet decomposition/reconstruction tool implemented in the Matlab "Wavelet Toolbox" to calculate the average spacing between boundary peaks for individual nucleosomes.

This parameter (which was estimated for every independent experiment) defined the distance that the individual profiles from each DNA strand had to be shifted to converge and define the midpoint position of each nucleosome. The resulting combined profile was wavelet-smoothed to generate the final nucleosome positioning profile. The sequencing coverage for every nucleotide was divided by the average genomic coverage to normalize the different experiments. We defined nucleosome-depleted regions (NDRs) as regions spanning at least 150 nucleotides (corresponding to the eviction of at least one nucleosome) with a normalized sequence coverage lower than 0.4.

## ChIP-seq library preparation and analyses

ChIP-Seq was performed as described in our previous work[26,52]. ChIP was performed as for the ChIP-qPCR experiments, with the exception that for the Rpc1-TAP ChIP-seq assays, an equal amount (10%) of *S. cerevisiae* Rpb3-TAP strain were mixed into the wild-type and mutant strains before the crosslink for spiked-in normalization. Similarly, for the Rpb3-HA ChIP-seq assays, an equal amount (10%) of *S. cerevisiae* Rpb3-HA strain were mixed in the *S. pombe* samples before the crosslink. High-throughput sequencing libraries were generated from 10 ng of the ChIP DNA using the Illumina TruSeq ChIP-seq library preparation protocol according to the manufacturer's instructions. The libraries were sequenced on an Illumina HiSeq 2500 apparatus.

For the analysis, raw reads were trimmed using trimmomatic against the appropriate adapter sequences with options SE ILLUMINA-CLIP:$adapters:2:30:10 LEADING:3 TRAILING:3 SLIDINGWINDOW:4:15 MINLEN:36. Trimmed reads were then mapped on the concatenation of the *S. pombe* genome (version ASM294v2.26) with the *S. cerevisaie* reference genome using HISAT2 with option–no-splice-alignment. Normalization factors were computed based on the number of spike-in reads mapped to the *S. cerevisiae* genome. Based on these normalization factors, we used deepTools to normalized coverage files (.bigwig) with options bamCoverage --samFlagExclude 256 --maxFragmentLength --scaleFactor $size_factor -bs 1 -of bigwig. Raw reads and processed genome coverage files in bigWig format are provided and available on GEO.

## NET-seq

NET-Seq libraries were constructed from biological duplicates of *rpb3-flag* cells and sequenced as described in our previous work[27]. Shortly, Libraries were constructed starting from 1 L of exponentially growing cells. RNAPII was immunoprecipitated using anti-FLAG M2 affinity gel, then recovered upon two successive elution steps using FLAG peptide. Ligation of DNA 3′-linker was performed from 2–3 μg of purified nascent RNA (i.e., co-immunoprecipitated with RNAPII; IP samples) and total RNA (input samples). Ligated RNAs were then submitted to alkaline fragmentation for 20 min at 95 °C. Single-end sequencing (50 nt) of the libraries was performed on a HiSeq 2500 sequencer, giving between 20 and 50 million reads. After trimming of the 5′-adapter, reads were uniquely mapped to the reference genome (ASM294v2.30) with a tolerance of two mismatches, giving 7.1–9.8 million uniquely mapped reads.

Data were normalized on the total number of uniquely mapped reads, for each sample. As a reference for the absence of transcription, we used a set of 50 protein-coding genes, showing the lowest nascent transcription signals (IP samples) and low total RNA levels (input samples). These genes were selected after filtering out the genes for which the signal was equal or close to zero as no (or very few) reads could be uniquely mapped, due to the high content of repetitive sequences. This was manually checked using the NET-seq signals (IP and input) and also the ChIP-seq signals (IP) for histone H3.

## RNA extraction

RNA was extracted as previously reported[53]. Briefly, 50 mL of *S. pombe* (OD$_{595nm}$ ~0.5) were harvested by centrifugation and washed in DEPC water. The cell pellets were resuspended in 750 μL of TES buffer

(10 mM Tris pH 7.5, 10 mM EDTA pH 8, 0.5% SDS) and the RNA was isolated by hot phenol extraction; equal volume of phenol:chloroform 5:1 was added (Sigma P1944), heated and shaken at 65 °C for 60 min, the upper phase was collected after centrifugation, mixed with an equal volume of phenol:chloroform:IAA 125:24:1 (Sigma 77619); the upper phase was then collected after centrifugation, mixed with an equal volume of chloroform:IAA 24:1 (Sigma 25666); the upper phase was collected again and the RNA was alcohol-precipitated with 2.5 volumes of 100% ethanol and 0.1 volume of 3 M sodium acetate pH 5.2 for minimum 30 min at −80 °C or overnight at −20 °C. After centrifugation, the pellet was washed with 70% ethanol, air-dried and resuspended in DEPC (Sigma D5758) water or pure water (Sigma 900682).

### RT-qPCR
In all, 0.5 μg of RNeasy-purified total RNA (Qiagen 74104) was retro-transcribed with the High-Capacity cDNA Reverse Transcription Kit (Thermo 4368813) by following the manufacturer's instructions. The real-time PCR amplification was performed with SYBR Green Supermix (Bio-Rad 172-5124) in a Bio-Rad CFX96TM Real-Time machine. The PCR program was 3 min at 95 °C and 40 cycles of (15 s at 95 °C and 30 s at 60 °C). Relative RNA quantification relied on the ΔΔCT method. Primer sequences can be found in Supplementary Data 1.

### Western blotting
Proteins were extracted as previously reported[54] from 10 mL of yeasts cultured in YES at 32 °C. Cells were pelleted, washed in water and disrupted with 0.3 M hydroxide sodium for 10 min at RT. The lysate was then centrifuged and the supernatant was discarded. The pellet was resuspended in 70 μL of alkaline extraction buffer (60 mM Tris-HCl pH 6.8, 4% β- mercaptoethanol, 4% SDS, 0.01% bromophenol blue, 5% glycerol) and boiled for 5 min at 95 °C. Overall, 10 μL of the samples were loaded on a 4–15% Mini-PROTEAN TGX Precast Protein Gel (Bio-Rad 456-1083), transferred on a nitrocellulose membrane (Bio-Rad 1704158) and blocked for 60 min at RT or overnight at 4 °C in 50:50 1× PBS:Odyssey Blocking Buffer PBS (Westburg LI 927-40100). The membrane was incubated with the primary antibody for 60 min at RT or overnight at 4 °C with Odyssey Buffer containing 0.05% Tween20 (Bio-Rad 161-0781) and a 1:500−1000 dilution of anti-α-tubulin (Sigma T5168) or anti-Histone H3 (Abcam ab1791) primary antibodies. After three washes in PBS-T, the membrane was incubated with a 1:10,000 dilution of anti-mouse (Westburg LI 925-32210) or anti-rabbit (Westburg LI 925-32211) secondary antibodies for 60 min at RT in Odyssey Buffer containing 0.05% Tween20. After three washes in PBS-T and three washes in PBS, the membrane was dried for 60 min at 37 °C and visualized on a LiCOR scanner on channel 800. The quantification was performed with ImageJ[55].

### RNA-sequencing and analyses
In total, 1 μg of total RNA was ribodepleted as described in the Rho-seq section and analyzed by microchip electrophoresis as described above. Overall, 125 ng of ribodepleted samples were prepared for strand-specific NGS with the TruSeq Stranded Total RNA Sample Preparation Protocol (Illumina). The library was sequenced on Illumina HiSeq 2500 (paired-end reads, 2× 50 nt). Reads were mapped on the genome using TopHat2. The quantification was made with feature-Counts function on R and the differential expression analysis was performed with DESeq2. The genes following these criteria were considered as differentially expressed; false discovery rate <10% and the absolute value of the fold change >1.5.

### Statistics and reproducibility
No statistical method was used to predetermine sample size. No data were excluded from the analyses. The experiments were not randomized. The Investigators were not blinded to allocation during experiments and outcome assessment. For all experiments with statistical analyses, the averaged value from three biological replicates is shown and error bars are the standard deviation. Upon paired *t* test, the obtained *P* value is indicated in the Source data file and represented in the figure as: \**P* < 0.05, \*\**P* < 0.01, \*\*\**P* < 0.001; ns not significant. When normalized data are shown (For CTD S2P and H3K14ac level), the independent data and error bars are provided in the Source data file and the normalized data is shown in the figures.

### Reporting summary
Further information on research design is available in the Nature Portfolio Reporting Summary linked to this article.

## Data availability
MNase-seq data[25] are available under the NCBI Gene Expression Omnibus (GEO) accession number GSE59768. NET-seq data[27] are available under the GEO accession number GSE72382. Pol III ChIP-seq and RNA-seq data in WT and rpb1 CTD S2A (this study) are available on GEO under the accession number GSE193552. The data and materials including strains presented in this study are available from the corresponding author upon request. Source data are provided with this paper.

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

## Acknowledgements

We are grateful to Elena Hidalgo, Marc Buhler, Sigurd Braun, Vincent Vanoosthuyse and Dom Helmlinger for reagents. We thank Odil Porrua, and Domenico Libri for comments on the manuscript. High-throughput sequencing was performed by the NGS platform of Institut Curie,

supported by the grants ANR-10-EQPX-03 and ANR-10-INBS-09-08 from the Agence Nationale de la Recherche (investissements d'avenir) and by the Canceropôle Ile-de-France. This work was also supported by ERC regulncRNA and ANR DNAlife to A.M., PDR T.0012.14, CDR J.0066.16, PDR T.0112.21 to D.H. C.Y.-S. is a FNRS Postdoctoral Researcher. D.H. is a FNRS Directors of Research.

## Author contributions

C.Y.-S. and V.M. performed the experiments. M.W. and A.M. contributed to the design and funding of the high-throughput analyses. M.L. and F.B. contributed the Rpb3-HA ChIP-seq experiment. D.H. and C.Y.-S. designed the project. D.H. acquired funding, supervised the experiments and wrote the paper with inputs from all authors.

## Competing interests

The authors declare no competing interests.
