## [Peer Review File · Nature Communications]

Chromatin remodeling by Pol II primes efficient Pol III transcriptionREVIEWER COMMENTS

Reviewer #1 (Remarks to the Author):

The authors show that S2 phosphorylation of Pol II CTD is required for the maintenance of nucleosome depleted region (NDR) in tRNA genes. Loss of NDR in S2Pmutant/S2Pdepletion is correlated with lower Pol III occupancy. By exploring previously published high through data they found S2 phosphorylated Pol II to be associated with virtually all tRNA loci together with low abundance of nascent Pol II transcripts mapping antisense to tRNA. Finally, they confirmed S2P-dependent Pol II occupancy of tRNA genes by ChIP. Further experiments showed that the SAGA-associated chromatin remodeling and the Mst2-associated histone acetylation are required for efficient Pol III transcription. The manuscript is clearly written and the figures are well prepared. The novelty of results and methods deserves its publication in Nature Communications.

Notwithstanding the general soundness of the study reported in the manuscript, there are a few issues/conclusions that I did not find fully convincing or satisfactorily developed.

The following points should be clarified in further versions of the manuscript.

1. In my opinion, the main point of novelty of this study is the discovery of Pol II antisense transcription at Pol III genes. However, the documentation should be strengthened by additional data. Netseq data (Fig. 3C) should be supplemented by showing the sense strand on aligned tRNA genes. The documentation of the presence of an exosome sensitive RNA antisense to tRNA is not convincing (Fig. S1A). It should be strengthened either experimentally by the creation of Pol II sequencing profiles in an exosome mutant or, if possible, by the analysis of previously reported data.
2. The transcription of selected Pol III genes analyzed by Rpc1 ChIP (presented on Fig. 2C) should be determined by Northern blotting with probes specific to tRNA precursors. Lack of change in tRNA levels in rpb1 CTD S2A mutant despite lower Pol III occupancy is confusing. It is unclear how tRNA levels were determined. Exactly which RNA seq experiment is referred to in the text (page 8)? Data are not presented because Fig. S3 is not included in the submission (fail Fig. S3 shows Fig.3).
3. The left panel on Fig. 4A shows the effect of pcr1 deletion mutation on Pol II association with several Pol III genes. It should be clarified whether the Pcr1 transcription factor participates in the recruitment of Pol II to all Pol III genes or to the subclass of Pol III genes? This issue should be experimentally addressed, ideally through the analysis of the ChIPseq data for pcr1 deletion mutant, or, at least, discussed.
4. Fig. 4A, middle panel, suggests direct contribution of Pcr1 to the maintenance of nucleosome depletion at Pol III loci. Is the function of Pcr1 dependent on Pol II phosphorylation of CTD at S2? Such a possibility should be experimentally studied by addressing the synergistic effects of Pcr1 deletion and S2P depletion on H3 levels on Pol III genes.
5. Both Pol III and Pol II occupancy on tRNA genes are dependent on S2 phosphorylation in CTD. What is missing is a general comment on how Pol III association with particular tRNA genes is correlated with Pol II association.

Minor comments:

1. The statement from the abstract that a Pol II-mediated mechanism enables efficient Pol III recruitment upon re-initiation of growth from the stationary phase is an overestimation. This is a new mechanism which contributes to Pol III activation together with others (e.g. Maf1 phosphorylation).
2. There is a mistake in the title of Figure 1; the results of Pol III occupancy experiments are not included in Fig. 1 but in Fig. 2.
3. The reference to figure 4A at the bottom of page 8 is wrong
4. Introduction: The phrase concerning TFIIIIB "Which itself requires TFIIIC...." Should be replaced by "Which itself is recruited by TFIIIC....." because in the in vitro system TFIIIC is only required for Pol III initiation and further steps need TFIIIIB and Pol III only.
5. The statement in the introduction: "...how RSC is targeted to these loci remains unknown" needs a comment. The link is provided by the Rsc4 subunit of RSC, which interacts with Rpb5, a subunit shared by three RNA polymerases (Soutourina et al., MCB 2006).
6. The statement in the introduction "repressing Pol III by nutritional starvation does not result in a

local gain of nucleosome (Parnell et al., 2008), which suggests that the remodeling of chromatin may be independent of high level of Pol III" needs a comment. Results by Bhargava and Arimbasseri, MCB 2008 show that chromatin structure directly participates in the regulation of a Pol III transcribed gene under different states of its activity in vivo.

7. The role of Pol III genes in the organization of chromatin should be mentioned in the introduction or in the discussion. 52 out of 174 fission yeast tRNA genes are located at centromeres and some centromeric tRNA genes have been shown to function as a heterochromatin barrier (Noma et al., 2006; Scott et al., 2006, 2007). Further analysis reveals a global chromosome organization by which dispersed tRNA and 5S rRNA genes frequently localize in proximity to centromeres (Iwasaki et al., 2010). Furthermore, TFIIC bounding to tRNA genes act as boundary elements that restrict the spreading of heterochromatin. This mechanism is linked to H3 methylation (Liu et al, NAR 2015).

Reviewer #2 (Remarks to the Author):

Overall, this is an excellent study. Although there are indications that pol II cooperates with pol III to regulate the transcription of tDNAs and other class III genes, this relationship is still not well understood. The authors propose that pol II transcription induces chromatin remodelling through the activity of the Gcn5/SAGA complex and Mst2. This is an interesting model, well supported by the data. The model is the fission yeast *S. pombe*, an excellent model for the study of chromatin regulation, but the findings are interesting and relevant beyond yeast researchers.

1. The only major point is the lack of use of statistics throughout the paper. This
2. Figure 1A. Expand name of mutant strain (not just S2)
3. Figure 1B. There is something strange with the y-axis label
4. Figure 1D. Decimal points should be points, not commas (same in figure 3D).
5. Introduce *rpc1* the first time it is used (as *rpc1-TAP*) to help the non-specialist
6. Figures 3A, 3B. The colours in the figure legend and in the actual legend are inconsistent (i.e., one is wrong)
7. Figure s3 legend – I don't understand what is plotted – would a scatter plot be simpler?
8. Figure 6B. The assays of cell growth / variability are not very convincing. I would suggest plating a fixed number of cells on a plate and measuring the number of colony-forming units, and to measure growth rates using optical densities.
9. The model figure is nice, I would put it in the main text if space allows.

Reviewer #3 (Remarks to the Author):

The manuscript by Carlo et al. investigate the transcriptional regulations of Pol III by Pol II. The study is based on a combination of ChIP-seq, quantitative PCR, genetic mutation, and reanalyzing of the publicly available dataset. The authors tried to show that Pol II facilitates the recruitment of histone acetyltransferases Gcn5 and Mst2 to tRNA genes by SAGA and Ser2P Pol II CTD, which is critical for the initiated growth from the stationary phase in yeast.

The manuscript relies heavily on correlations, and most of the conclusions are overstated. Additionally, numerous controls were not included, and no statistical analyses were performed. This makes any conclusions open to interpretation and vague. The authors proposed a model for the regulation of Pol III transcription, but few transcription-related assays were carried out, and the mature tRNAs do not decrease in CTD S2A mutant. Thus, the current manuscript is quite preliminary and not ready for publication. Specific concerns are listed below.

Figure 1.

RPB1 CTD S2A mutant and Ser2P inhibition were used to investigate the role of Ser2P in nucleosome positioning at tDNA sites. The authors did not investigate their effects on total Pol II or Ser5P Pol II occupancy. The Ser5P inhibitor THZ1 and initiation inhibitors were also not examined. These missed controls are essential to determine whether the effects observed by the authors are Ser2P specific or not.

Figure 2.

RPC1 ChIP-seq and qPCR were used to examine the effects of S2A mutant. The tDNA loci or Pol III occupied non-tRNA peaks, without Pol II binding, should be shown as a negative control for ChIP-seq and qPCR analyses, which would help the authors to determine whether these effects are specific for tRNAs or just nonspecific secondary effects.

Figure 3.

Observing Ser2P and Ser5P signals at tRNA loci is interesting, but the reviewers are not convinced because these signals may be the background. The authors should show Ser2P and Ser5P signals at mRNA regions and ChIP-seq input signals as controls to confirm that the tDNA loci-associated Ser2P and Ser5P signals are not background signals. This is the same for the Pol II NET-seq. The NET-seq signals at mRNA regions need to be shown for comparison as well. In Figure 3D, the author indeed showed that Ser2P inhibition decreased the chromatin binding of RPB3, which supports the reviewer's worry about the decrease of Pol II occupancy instead of Ser2P causing the effects on Pol III.

Figure 4.

The non-responsive genes were not shown for all these qPCR experiments. Since *pcr1* deletion also caused the decrease of Pol II occupancy and the decrease of Pol III binding, which are consistent with the recent finding of decreased Pol III occupancy after acute Pol II degradation. Their results make the reviewer further concerned the conclusion related to Ser2P could also be explained by the total Pol II occupancy.

The effects of Gcn5 on histone acetylation are expected, which cannot support the author's statement that "SAGA-associated chromatin remodeling is required for efficient Pol III transcription". The effects of active transcription, such as TT-seq-related nascent transcript analyses, were not performed at all.

Figure 5.

Only Mst2 and histone 3 occupancies were investigated. The Pol III transcription was not examined. The authors only investigated the effects of deletion of Mst2 and Gcn5, the functions of histone acetylation were also not investigated. The experimental evidence does not well support the conclusions, and the statement by the authors is far-fetched. Genomics data of Pol III, Pol II, and nascent RNA analyses should be produced for these useful mutants to provide more quantitative and reliable evidence.

Figure 6.

The effects of S2A mutant are similar to the effects after deletions of *gcn5*, and *mst2* does not mean they are connected. The insertion of the tRNA terminator only examined the Pol III occupancy, and the physiological function was not examined.

Minor:

1. The error bars for qPCR of H3K4ac in Figure 4B and CTD S2P in Figure 3D and Figure S4B are missing.
2. The CRE motif is found within 1 kb downstream of 30% of the tDNA, as reported, and Ser2P, Ser5P, and Pol II peaks are shifted upstream or almost overlay with tDNA. How to explain this conflict?
3. Some of their model genes did not behave similarly or robustly in different panels, such as *snu6* in Figure 4B, *THR.10* in Figure 5A, *ARG.10* in Figure 6A.

4. Their sequencing-related methods are too brief. More experimental details should be provided. The bioinformatics analysis methods should also provide.

Response to the referees

Reviewers' Comments:

Reviewer#1

The manuscript is clearly written and the figures are well prepared. The novelty of results and methods deserves its publication in Nature Communications.

Notwithstanding the general soundness of the study reported in the manuscript, there are a few issues/conclusions that I did not find fully convincing or satisfactorily developed.

We thank the referee for his/her comments and we are happy to hear that he/she believes that our work deserves publication in *Nature Communications*. We have responded to the issues raised below.

1-Nesteq data (Fig. 3C) should be supplemented by showing the sense strand on aligned tRNA genes.

We have added as Figure 3D the average profiles of sense and antisense Netseq data, which shows that the Rpb3-flag specific signal is mostly antisense. Note that sense tRNA signal in the input arises from non-specific interactions of tRNAs with the beads during NET-seq library preparation, as described in the original Netseq paper (Churchman and Weissman, 2011).

- The documentation of the presence of an exosome sensitive RNA antisense to tRNA is not convincing (Fig. S1A). It should be strengthened either experimentally by the creation of Pol II sequencing profiles in an exosome mutant or, if possible, by the analysis of previously reported data.

As suggested by the referee, we have analyzed previous dataset from Watts et al., 2018 (GSE104713). Figures S1B confirms that the inactivation of the exosome in the *rrp6* mutant results in the accumulation of both sense and antisense transcripts arising from tDNA genes while protein coding genes do not show a similar trend.

2-The transcription of selected Pol III genes analyzed by Rpc1 ChIP (presented on Fig. 2C) should be determined by Northern blotting with probes specific to tRNA precursors. Lack of change in tRNA levels in *rpb1* CTD S2A mutant despite lower Pol III occupancy is confusing. It is unclear how tRNA levels were determined. Exactly which RNA seq experiment is referred to in the text (page 8)? Data are not presented because Fig. S3 is not included in the submission (fail Fig. S3 shows Fig.3).

We now present in Figure S3 both a Rpc1-TAP ChIP and a northern blot analysis of tRNA ARG.05 in *wt*, *rpc25-flag* and *rpb1 S2A* or the combined *rpc25-flag rpb1 S2A* backgrounds. This experiment confirms the disconnect between the level of Pol III determined by Rpc1-TAP ChIP and the level of the mature tRNAs that is unaffected. It appears that the level of precursors is slightly increased in the *rpb1 S2A* strain while decreased in the *rpc25-flag* strain. In the double mutant, wt levels of both precursors and mature is observed, although the level of Rpc1-TAP is still reduced. These data support that a post-transcriptional buffering system is active. We apologize for the absence of Figure S3 in the first submission.

3- The left panel on Fig. 4A shows the effect of *pcr1* deletion mutation on Pol II association with several Pol III genes. It should be clarified whether the Pcr1 transcription factor participates in the recruitment of Pol II to all Pol III genes or to the subclass of Pol III genes? This issue should be experimentally addressed, ideally through the analysis of the ChIPseq data for *pcr1* deletion mutant, or, at least, discussed.

To address this issue, we have performed a Rpb3-HA ChIP-seq experiment in the *pcr1* deletion strain, as suggested by the referee. This experiment revealed that about 44% of the tDNA loci have a reduced level of Pol II when *pcr1* is absent. We now discuss this data as follows:

"At about 44% of the tDNA loci, the Pol II transcription we detected at a subset of class III loci requires the Pcr1 transcription factor that was shown to be enriched at tRNA genes¹. Other transcription factors, including Gaf1 may participate in the recruitment of Pol II²". (Page 14)

4- Fig. 4A, middle panel, suggests direct contribution of Pcr1 to the maintenance of nucleosome depletion at Pol III loci. Is the function of Pcr1 dependent on Pol II phosphorylation of CTD at S2? Such a possibility should be experimentally studied by addressing the synergistic effects of Pcr1 deletion and S2P depletion on H3 levels on Pol III genes.

To address this issue, we have constructed a *pcr1Δ rpb1 S2A* double mutant and repeated the H3 ChIP experiment of Figure 4 (middle panel). As shown below, no synergistic effect is observed, supporting that the function of Pcr1 is likely dependent on Pol II phosphorylation.

5- Both Pol III and Pol II occupancy on tRNA genes are dependent on S2 phosphorylation in CTD. What is missing is a general comment on how Pol III association with particular tRNA genes is correlated with Pol II association.

We have not found significant correlations between the level of Rpc1 (determined by ChIP-seq) and the level of antisense Pol II transcription (determined by NET-seq), suggesting that Pol II transcription is not limiting. It should be noted in that context that the expression of class III genes is less complex in yeast compared to higher eukaryotes: all class III genes are transcribed with similar level of Pol III and Pol II is detected at all these loci.

This is now summarized in the discussion as:

“As we did not find significant correlation between the level of Pol III and Pol II, Pol II transcription may not be limiting for efficient Pol III occupancy, at least in exponential growth.” (Page 7)

Minor comments:

1. The statement from the abstract that a Pol II-mediated mechanism enables efficient Pol III recruitment upon re-initiation of growth from the stationary phase is an overestimation. This is a new mechanism which contributes to Pol III activation together with others (e.g. Maf1 phosphorylation).

The sentence from the abstract was modified as follows:

We report here a new mechanism where RNA Polymerase II (Pol II) transcription is required to prime and maintain nucleosome depletion at Pol III loci and contributes to efficient Pol III recruitment upon re-initiation of growth from stationary phase in Fission yeast.

2. There is a mistake in the title of Figure 1; the results of Pol III occupancy experiments are not included in Fig. 1 but in Fig. 2.

The title was modified accordingly. We apologize for this mistake.

3. The reference to figure 4A at the bottom of page 8 is wrong

The reference to Figure 4A was modified accordingly.

4. Introduction: The phrase concerning TFIIIB “Which itself requires TFIIIC....” Should be replaced by “Which itself is recruited by TFIIIC.....” because in the in vitro system TFIIIC is only required for Pol III initiation and further steps need TFIIIB and Pol III only.

The sentence was modified accordingly.

5. The statement in the introduction: “...how RSC is targeted to these loci remains unknown” needs a comment. The link is provided by the Rsc4 subunit of RSC, which interacts with Rpb5, a subunit shared by three RNA polymerases (Soutourina et al., MCB 2006).

We have added this reference as follows:

“These data support that RSC is required to maintain class III genes in a low nucleosome density favorable to Pol III transcription although how RSC is targeted to these loci remains unclear although a physical interaction between Rsc4 C-terminus and Rpb5, a shared subunit of the three polymerases, was reported (Soutourina et al., 2006).” (Page 4).

6. The statement in the introduction “repressing Pol III by nutritional starvation does not result in a local gain of nucleosome (Parnell et al., 2008), which suggests that the remodeling of chromatin may be independent of high level of Pol III” needs a comment. Results by Bhargava and Arimbasseri, MCB 2008 show that chromatin structure directly participates in the regulation of a Pol III transcribed gene under different states of its activity in vivo.

To take this reference into account, we have added the following sentence in the discussion:

“Nevertheless, in the case of the SNR6 gene, the upstream nucleosome, which covers the TATA box under repressed conditions is shifted approximately 50 bp further upstream by RSC upon activation (Arimbasseri et al., 2008).” (Page 8).

7. The role of Pol III genes in the organization of chromatin should be mentioned in the introduction or in the discussion. 52 out of 174 fission yeast tRNA genes are located at centromeres and some centromeric tRNA genes have been shown to function as a heterochromatin barrier (Noma et al., 2006; Scott et al., 2006, 2007). Further analysis reveals a global chromosome organization by which dispersed tRNA and 5S rRNA genes frequently localize in proximity to centromeres (Iwasaki et al., 2010). Furthermore, TFIIC bounding to tRNA genes act as boundary elements that restrict the spreading of heterochromatin. This mechanism is linked to H3 methylation (Liu et al, NAR 2015).

While we are aware of the role of class III genes in limiting heterochromatin spreading in fission yeast, we believe that this aspect is not directly related to our own work and that the proposed references, though interesting on their own may unfocus the study. However, if the referee believes that they are essential to the paper, we will add them.

We thank the referee for its numerous suggestions that improve the quality of our paper.

Reviewer#2

We were happy to read that the referee believes that overall our work is an excellent study and that the model presented is well supported by the data.

1. The only major point is the lack of use of statistics throughout the paper.

We have addressed this issue by presenting t-Test data and associated p-values when relevant in all figures.

2. Figure 1A. Expand name of mutant strain (not just S2)

This is now corrected accordingly

3. Figure 1B. There is something strange with the y-axis label

This is now corrected accordingly

4. Figure 1D. Decimal points should be points, not commas (same in figure 3D).

This is now corrected accordingly

5. Introduce rpc1 the first time it is used (as rpc1-TAP) to help the non-specialist

This is now corrected accordingly

6. Figures 3A, 3B. The colours in the figure legend and in the actual legend are inconsistent (i.e., one is wrong)

This is now corrected accordingly

7. Figure s3 legend – I don't understand what is plotted – would a scatter plot be simpler?

This figure is now modified and shows a Boxplot of the log₂ fold-change for pre- or mature tRNAs (n = 171) from RNA-seq in the indicated strains.

8. Figure 6B. The assays of cell growth / variability are not very convincing. I would suggest plating a fixed number of cells on a plate and measuring the number of colony-forming units, and to measure growth rates using optical densities.

We now present both drop assays with additional conditions (longer starvation before plating) in Figure 6B and, as suggested, OD-based growth rates in Figure S5A. Perhaps as expected, the use of longer starvation time increased the biological effect on growth, which we think strengthens our data.

9. The model figure is nice, I would put it in the main text if space allows.

Unfortunately, there is no room left in the main figures, so we have to keep the model in supplementary files.

Reviewer #3

The manuscript relies heavily on correlations, and most of the conclusions are overstated. Additionally, numerous controls were not included, and no statistical analyses were performed. This makes any conclusions open to interpretation and vague. The authors proposed a model for the regulation of Pol III transcription, but few transcription-related assays were carried out, and the mature tRNAs do not decrease in CTD S2A mutant. Thus, the current manuscript is quite preliminary and not ready for publication. Specific concerns are listed below.

Even though the overall opinion of this referee differs from the other two referees and is not supportive, we thank him/her for his/her thorough review and the constructive nature of his/her comments, which we have been able to largely address. Specifically, the referee states that numerous controls are missing and that statistical analyses were not performed (an issue shared by referee 2). In addition, the referee indicates that few transcription-related assays were carried out and that the mature tRNAs do not decrease in the CTD S2A mutant. Finally, the referee is worried that acute Pol II degradation might occur in the *rpb1* CTD S2A mutant, which could lead to the observed Pol III decrease (as described recently in a different system). We have now performed additional experiments and added additional controls (including a control against gross Pol II degradation/dissociation from chromatin in the CTD S2A), which we hope will convince this referee of the soundness of our study.

Figure 1.

RPB1 CTD S2A mutant and Ser2P inhibition were used to investigate the role of Ser2P in nucleosome positioning at tDNA sites. The authors did not investigate their effects on total Pol II or Ser5P Pol II occupancy. The Ser5P inhibitor THZ1 and initiation inhibitors were also not examined. These missed controls are essential to determine whether the effects observed by the authors are Ser2P specific or not.

To address the concern of the effect of the *rpb1* CTD S2A mutant and Ser2P inhibition on total Pol II, we have performed Rpb-3HA ChIP-seq in the wt and the CTD S2A mutant, which reveals that the CTD S2A mutation does not have strong effect on the global Pol II occupancy at either protein coding genes (Figure 1E top) or tDNA (Figure 1E bottom). This result is in line with previous works from us and others showing that in fission yeast, the transcriptome of the *rpb1* CTD S2A mutant is very similar to that of a wild-type, with the exception of gene-specific defects (Coudreuse et al, 2010).

Treatment with the CTD Ser5P inhibitor THZ1 was reported to also sharply lower CTD Ser2P (Kwiatkowski et al., 2014, <https://doi.org/10.1038/nature13393>), which compromises its use as a Ser5-specific perturbation. Similarly, using inhibitors of initiation is expected to broadly affect transcription, which would render any conclusions about an effect on tDNA speculative. Moreover, to the best of our knowledge, none of these inhibitors were previously used in fission yeast. Nevertheless, we have used an analogue-sensitive version of the CTD S5 kinase Mcs6-as (previously fully described in Devos et al., 2015). The fast decrease of CTD S5P resulting from Mcs6-as inhibition does not have similar effects than the depletion of CTD S2P on H3 level at tDNA. These data show that the effect we observe when CTD S2P is depleted is specific of this mark and does not result from a global deregulation of Pol II transcription.

Figure 2.

RPC1 ChIP-seq and qPCR were used to examine the effects of S2A mutant. The tDNA loci or Pol III occupied non-tRNA peaks, without Pol II binding, should be shown as a negative control for ChIP-seq and qPCR analyses, which would help the authors to determine whether these effects are specific for tRNAs or just nonspecific secondary effects.

The referee proposes to use Pol III occupied non-tRNA peaks, without Pol II binding as a negative control for ChIP-seq and qPCR analyses. However, we did not find any loci displaying these combined features. However,

we would like to point out that the input values are shown in Figure 2A (ChIP-seq) and the IP signals are clearly peaking within a region covering the tDNA loci. There is therefore little doubt that the Rpc1-TAP signal is specific to these loci. In addition, in Figure 2C an untagged control is presented for each locus, again clearly showing that the Rpc1-TAP signal is specific. We also reproduce hereafter two data previously published in Materne et al, 2015 clearly showing that nucleosome occupancy is not globally affected in the *rpb1 CTD S2A* mutant, especially at the NDR (nucleosome depleted region) found close to the TSS (Transcription start site) and TTS (Transcription termination site).

Left panel: Meta-gene analysis of the nucleosome occupancy signal for all protein-coding genes near the TTS. The distance between the TSS and the average -1 nucleosome midpoint position is indicated in blue for the wt and red for the *rpb1 CTD S2A* mutant.

Right panel: Meta-gene analysis of the nucleosome occupancy signal for all protein coding genes near the TSS. The distance between the TTS and the average last nucleosome midpoint position is indicated in blue for the wt and red for the *rpb1 CTD S2A* mutant.

The massive increase in occupancy observed in Figure 1B (current manuscript) for tDNA, the effect appears very specific for tDNA.

Figure 3.

Observing Ser2P and Ser5P signals at tRNA loci is interesting, but the reviewers are not convinced because these signals may be the background. The authors should show Ser2P and Ser5P signals at mRNA regions and ChIP-seq input signals as controls to confirm that the tDNA loci-associated Ser2P and Ser5P signals are not background signals.

First, we invite the referee to look at Figure 3A and 3B where Pol II, S2P and S5P are shown over an 800 bp region covering the tDNA loci. It is very clear that an enrichment is observed around the tRNA TSS compared to the flanking regions. The additional genome-wide data we present in Figure 1E to respond to the first concern of the referee (see Figure 1 above) also show a clear and directional enrichment of Pol II at tDNA compared to flanking 1 kb regions (Figure 1E bottom). A comparison with protein coding genes (Figure 1E top) is also shown, as requested by the referee. Importantly, these data were spike-in normalized.

We also would like to point out that several studies in other systems have also detected a Pol II enrichment at tDNA loci³⁻⁵.

This is the same for the Pol II NET-seq. The NET-seq signals at mRNA regions need to be shown for comparison as well.

We have previously published all the NET-seq data related to class II genes in Wery et al, RNA 2017.

We now show the input and IP signals for sense and antisense transcription at tDNA loci. The massive input sense signal corresponds to mature tRNAs (resulting from Pol III transcription) that are known contaminants in NET-seq experiments (Churchman and Weissman, 2011). At the contrary, a clear antisense Pol II IP signal is detected over tDNA loci, strongly enriched from the input. These data indicate that the NET-seq signal detected is well over the background compared to 200 bp flanking regions.

In Figure 3D, the author indeed showed that Ser2P inhibition decreased the chromatin binding of RPB3, which supports the reviewer's worry about the decrease of Pol II occupancy instead of Ser2P causing the effects on Pol III.

We now present a genome-wide analysis of Pol II occupancy at tDNA and protein coding genes in wt and the CTD S2A mutant in Figure 1E. Note that as for our other comparative ChIP-seq experiments, spiked-in *S. cerevisiae* chromatin was used to control against potential global changes in Pol II occupancy. Consistently with previous literature showing that the CTD S2P in fission yeast is dispensable for both viability and the efficient expression of most genes (Coudreuse et al.), this metagene analysis shows little overall effect of the CTD S2A mutation on Pol II occupancy. It is therefore very unlikely that changes in Pol II occupancy rather than a specific decrease of Pol II CTD S2P are the cause of the effect on Pol III. The effect observed in Figure 3E when the S2 kinase is inhibited (30 minutes inhibition) may be very transient as it is not obvious in the CTD S2A strain.

Figure 4.

The non-responsive genes were not shown for all these qPCR experiments. Since pcr1 deletion also caused the decrease of Pol II occupancy and the decrease of Pol III binding, which are consistent with the recent finding of decreased Pol III occupancy after acute Pol II degradation. Their results make the reviewer further concerned the conclusion related to Ser2P could also be explained by the total Pol II occupancy.

As indicated above, the total Pol II occupancy is barely affected in the CTD S2A mutant and therefore cannot explain the effect on the chromatin template at class III loci.

The effects of Gcn5 on histone acetylation are expected, which cannot support the author's statement that "SAGA-associated chromatin remodeling is required for efficient Pol III transcription". The effects of active transcription, such as TT-seq-related nascent transcript analyses, were not performed at all.

We are surprised by the comment of the referee. We report the effect of Gcn5 on histone acetylation at tDNA. Even though this effect may be expected, it has not been demonstrated until now. Considering that the association of Gcn5 with SAGA (Helmlinger et al., 2008) is well-documented, we don't understand why there is an issue with our statement that "SAGA-associated chromatin remodeling is required for efficient Pol III transcription".

In addition, we report the presence at tDNA loci of Pol II by ChIP-seq and NET-seq, which reveals active transcription.

Figure 5.

Only Mst2 and histone 3 occupancies were investigated. The Pol III transcription was not examined. The authors only investigated the effects of deletion of Mst2 and Gcn5, the functions of histone acetylation were also not investigated. The experimental evidence does not well support the conclusions, and the statement by the authors is far-fetched. Genomics data of Pol III, Pol II, and nascent RNA analyses should be produced for these useful mutants to provide more quantitative and reliable evidence.

We are surprised to read the statement of the referee that: *The authors only investigated the effects of deletion of Mst2 and Gcn5*. Indeed, we show that:

- Figure 5 shows that Gcn5 is enriched at 7 tDNA loci and that its absence results in a decrease of H3K14 acetylation at these loci.
- Figure 6 shows that Mst2 is enriched at 7 tDNA loci, which requires Set2 and H3K36me3. In addition, the absence of *gcn5* and *mst2* synergistically increase H3 occupancy at these loci.

- Figure S4 shows that the absence of *gcn5* and *mst2* synergistically decrease H3K14 acetylation at these loci. In addition, this figure also shows that Rpc1-TAP occupancy is decreased on the *gcn5 mst2* double mutant.

To us, these data strongly support that these two histone acetyltransferases are physically associated with tDNAs loci and synergistically controls their level of acetylation. The referee concludes that “*these experimental evidence does not well support the conclusions, the statement by the authors is far-fetched*”. It is unclear to us why this conclusion is “*far-fetched*” and not reliable.

Figure 6.

The effects of S2A mutant are similar to the effects after deletions of *gcn5*, and *mst2* does not mean they are connected.

We agree with the referee. Besides, nowhere do we report a connection between *Gcn5* and the CTD S2P. Rather, *Gcn5* is likely recruited through the SAGA complex associated with Pol II. Regarding *Mst2*, we show that *Mst2* requires *Set2* and H3K36me3 to be recruited at tDNA loci. The connection between *Set2* and *Mst2* was previously studied in detail (Flury et al., 2017). In addition, there is a large body of evidence that *Set2* is directly recruited by the CTD S2P. Therefore, we believe that the CTD S2P-*Set2*-*Mst2* connection is strongly supported by experimental evidence and is expanded here to tDNA.

The insertion of the tRNA terminator only examined the Pol III occupancy, and the physiological function was not examined.

We have used the engineered tDNA *PRO.08* loci to directly analyze the effect of the absence of Pol II passage for efficient Pol III recruitment, which is clearly reported in Figure 6. The physiological function of this process is analyzed in Figure 6B showing that CTD S2P, *Gcn5* and *Mst2* are required to efficiently reach high level of Pol III occupancy upon exit from stationary phase (Figure 6A) and ensure growth (Figure 6B). If the referee refers to the physiological consequence of specifically impeding the expression of the *PRO.08* tRNA, there are 9 copies of the corresponding gene and we do not expect that perturbing the expression of this specific copy is going to dramatically affect cell growth.

Minor:

1. The error bars for qPCR of H3K4ac in Figure 4B and CTD S2P in Figure 3D and Figure S4B are missing.

The reason why error bars are not shown is that in both cases, the data shown results from a normalization (on total H3 for H3K4ac in Figure 4B and on total Pol II for CTD S2P in Figure 3D). It is therefore inappropriate to show the error bars. However, we can show the graphs for each individual data before normalization with error bars.

2. The CRE motif is found within 1 kb downstream of 30% of the tDNA, as reported, and Ser2P, Ser5P, and Pol II peaks are shifted upstream or almost overlay with tDNA. How to explain this conflict?

While the CRE motif is found within 1 kb, it is on average closer to the tDNA and the average Pol II ChIP profile shows that transcription starts mostly in a 200 to 300 bp window downstream of the tDNA.

3. Some of their model genes did not behave similarly or robustly in different panels, such as *snu6* in Figure 4B, *THR.10* in Figure 5A, *ARG.10* in Figure 6A.

These model genes were chosen randomly and we believe that it is expected that some behave a bit differently while the general trend is present. This likely depends on their chromatin and genomic neighborhood.

4. Their sequencing-related methods are too brief. More experimental details should be provided. The bioinformatics analysis methods should also provide.

All details are now added accordingly.

REVIEWERS' COMMENTS

Reviewer #1 (Remarks to the Author):

The revised manuscript by Carlo Yague-Sanz and others is vastly improved over prior version and represents a significant advance towards our understanding role of Pol II in control of Pol III transcription by chromatin remodeling.

The authors have put significant effort to improve the quality of the paper by performing additional experiments and analyses.

According to my suggestion the authors :

1. Have analyzed the average profiles of sense and antisense Netseq data and showed that the Rpb3-flag specific signal is mostly antisense (data included as new Fig. 3D)
2. analyzed previous dataset from Watts et al., 2018, by which confirmed that the inactivation of the exosome in the rrp6 mutant results in the accumulation of both sense and antisense transcripts arising from tDNA genes while protein coding genes do not show a similar trend. (new Fig. S1B)
3. added Fig. S3 in which they have shown that the level of precursors is slightly increased in the rpb1 S2A strain
4. have performed a new Rpb3-HA ChIP-seq experiment in the pcr1 deletion strain which revealed Pcr1 transcription factor to be enriched at tRNA genes (Fig. 4A)
5. constructed a pcr1D rpb1 S2A double mutant and repeated the H3 ChIP experiment supporting that the function of Pcr1 is likely dependent on Pol II phosphorylation.

My comments to the original version of the manuscript are sufficiently addressed and the main text contains additional explanation and information.

Since resubmitted manuscript resolves most of the issues that in my opinion existed in the previously submitted manuscript, I would recommend its publication in Nature Communications

Reviewer #2 (Remarks to the Author):

The authors have addressed all the points are made in my review

Reviewer #3 (Remarks to the Author):

My concerns have been addressed, thus, I support the publication of this work.

Response to the referees

Reviewers' Comments:

Reviewer#1

The revised manuscript by Carlo Yague-Sanz and others is vastly improved over prior version and represents a significant advance towards our understanding role of Pol II in control of Pol III transcription by chromatin remodeling.

The authors have put significant effort to improve the quality of the paper by performing additional experiments and analyses.

According to my suggestion the authors :

1. Have analyzed the average profiles of sense and antisense Netseq data and showed that the Rpb3-flag specific signal is mostly antisense (data included as new Fig. 3D)
2. analyzed previous dataset from Watts et al., 2018, by which confirmed that the inactivation of the exosome in the rrp6 mutant results in the accumulation of both sense and antisense transcripts arising from tDNA genes while protein coding genes do not show a similar trend. (new Fig. S1B)
3. added Fig. S3 in which they have shown that the level of precursors is slightly increased in the rpb1 S2A strain
4. have performed a new Rpb3-HA ChIP-seq experiment in the pcr1 deletion strain which revealed Pcr1 transcription factor to be enriched at tRNA genes (Fig. 4A)
5. constructed a pcr1D rpb1 S2A double mutant and repeated the H3 ChIP experiment supporting that the function of Pcr1 is likely dependent on Pol II phosphorylation.

My comments to the original version of the manuscript are sufficiently addressed and the main text contains additional explanation and information.

Since resubmitted manuscript resolves most of the issues that in my opinion existed in the previously submitted manuscript, I would recommend its publication in Nature Communications

We are happy to read that we have addressed the concerns raised by this referee.

Reviewer#2

The authors have addressed all the points are made in my review

We are happy to read that the referee is satisfied

Reviewer #3

My concerns have been addressed, thus, I support the publication of this work.

We are happy to read that the referee is satisfied